# Recent Progress in Identifying Bacteria with Fluorescent Probes

**DOI:** 10.3390/molecules27196440

**Published:** 2022-09-29

**Authors:** Yuefeng Ji, Guanhao Li, Juan Wang, Chunxiang Piao, Xin Zhou

**Affiliations:** 1Department of Food Science and Engineering, College of Integration Science, Yanbian University, Yanji 133002, China; 2Food Research Centre, Yanbian University, Yanji 133002, China; 3Department of Chemistry, College of Chemistry and Chemical Engineering, Qingdao University, Qingdao 266071, China

**Keywords:** bacteria, fluorescent probe, detection, imaging

## Abstract

The development of new techniques to rapidly and accurately detect bacteria has drawn continuous attention due to the potential threats posed by bacteria to human health and safety. Recently, a novel strategy based on fluorescent probes has drawn considerable interest for the detection of bacteria due to its high selectivity, fast response, and simple operation. In this review, we summarize the recent progress on fluorescent probes for the specific recognition and discrimination of Gram-negative and Gram-positive bacteria. In particular, we outline current design strategies, such as targeting of the differences in surface components, cell wall components, endogenous enzymes, surface charge, and hydrophobicity of various kinds of bacteria to develop various fluorescent sensors (organic small-molecule fluorescent probes, nanoprobes, and metal ion probes). We also emphasize the application of organic molecules in probe recognition elements. We hope that this review can stimulate this research area in bacterial detection and imaging in the future.

## 1. Introduction

The increasing public health problems caused by bacterial infections are the subject of continuous concern. In 2017, the World Health Organization (WHO) released a list of priority pathogens (Table 1), which are classified as critical, high, or medium according to their priority [1]. The spread of pathogens poses a major threat to morbidity and mortality around the world, and the emergence of antibiotic-resistant pathogens has undoubtedly accelerated this process. More than 2.8 million infections with antibiotic-resistant bacteria are reported in the United States each year, resulting in more than 35,000 deaths [2]. When the antibiotic pipeline is almost empty, the identification and quantification of pathogens are important for food safety, water environment analysis, and clinical diagnosis of public health protection [3].

Each kind of pathogenic bacteria is believed to differ from one another, for example, in terms of the type of surface of membrane phospholipids, various membrane proteins, and membrane charge. According to the composition and structure if the pathogen surface, bacteria can be divided into two categories: Gram-positive and Gram-negative [4,5]. Gram-positive bacteria cell walls have a thick peptidoglycan and prominent teichoic acid, with no outer membrane. Gram-negative bacteria have a relatively thinner peptidoglycan and are composed of phospholipid and lipopolysaccharide (LPS) lipoprotein outer membrane [6]. On the other hand, bacterial cell wall construction and basic biological metabolism depend on a variety of enzymes. For example, nitroreductase (NTR) and alkaline phosphatase are widely expressed in both Gram-positive and Gram-negative bacteria. In addition, antibiotic-resistant pathogens express high levels of *β*-lactamase used to destroy antibiotics [7,8,9].

Traditional bacterial detection methods include culture staining, microscope imaging, biochemical and metabolic analysis, and immunological methods. In recent years, molecular biological methods, biosensors, and mass spectrometry have been applied to the detection and differentiation of bacteria [10]. Among the numerous detection methods, fluorescence detection has the advantages of high sensitivity, simple operation, real-time imaging, and non-invasiveness and has become a powerful analysis tool to visualize the number and distribution of pathogenic bacteria [11].

A fluorescent probe is usually composed of three parts: a chromogenic element, a connecting element, and an identifying element. The chromogenic element plays an indispensable role in visualization and data collection in fluorescence detection. Chromogenic elements include organic dyes, quantum dots (QDs), conjugated polymers, etc. organic dyes, including fluorescein, rhodamine, cyanine, and alexa dyes, are often used as optical signals for optical biosensors [12]. QDs, also known as semiconductor nanoparticles, are quasi-zero-dimensional materials composed of tens to hundreds of atoms, typically with a size distribution between 2 and 10 nm. QDs are usually composed of groups II-VI (CdTe, CdSe, CdS, ZnSe, ZnS, and ZnTe), III-V (InAs and InP) and IV-VI (PdTe, PdSe, and PdS) in a two-element composition [13]. QDs have good optical stability, a wide excitation spectrum and narrow emission spectrum, and high Stokes displacement. The ability to determine the emission spectrum by changing the size of the QDs is also a unique advantage. The low toxicity of carbon QDs promotes the development of biological labeling and detection. Conjugated polymers are linearly conjugated polymers with π-π-conjugated electronic structures, such as polyacetylene, polyaniline, polyfluorene, and polythiophene. The ability of conjugated polymers to capture light and signal amplification effects has attracted attention in the field of bacterial detection [11,14,15].

The detection of bacteria by bacterial surfaces, cell walls, proteins, nucleic acids, enzymes, and metal elements has become increasingly popular [16,17]. Liu et al. first summarized the current application of fluorescent probes in bacterial cell wall structural response, and Huang et al. further reviewed the bacterial response sites on this basis, explaining the advantages of fluorescent probe detection and traditional detection methods [11,18]. In view of the spread of antibiotic resistance and the evolution of pathogenicity, the detection of drug-resistant bacteria by fluorescent probes is still challenging [19].

The target of the fluorescent probe is determined by different target sites of pathogenic bacteria. Herein, we summarize the currently available target recognition strategies of pathogen fluorescent probes, as documented in Figure 1: metabolic labeling of the surface components of pathogenic bacteria, identification of the cell wall components of pathogenic bacteria, binding to pathogenic endogenous enzymes, and nonspecific detection.

## 2. Metabolic Labeling of the Surface Components of Pathogenic Bacteria

Metabolic labeling is a technique for the incorporation of chemical molecules into macromolecules of interest via the intracellular enzyme. The combination of metabolic labeling and bio-orthogonal chemistry improves and extends traditional metabolic labeling strategies [20]. The probe in this method does not contain radioactive or stable isotopes but is characterized by chromogenic elements. Metabolic labeling is divided into direct and indirect labeling methods. In the direct labeling method, the fluorescent probes are constructed with fluorophore-modified biological orthogonal precursors and incubated with bacteria. The fluorescent probes are directly labeled on the surface of bacteria through the unique metabolic pathway. The indirect labeling method involves the use of metabolic precursors modified by biological orthogonal functional groups (such as alkyne and azide) to culture bacteria through click chemistry on the bacteria for fluorescence labeling [21,22]. The bacterial surface contains abundant metabolic compounds, among which peptidoglycan stem peptide, outer-membrane glycolipids, pseudoamino, and maltose are commonly used in metabolic labeling of pathogenic bacteria.

### 2.1. Peptidoglycan Stem Peptide

Peptidoglycan can be found in the cell walls of both Gram-positive and Gram-negative bacteria. Biological orthogonal labeling based on bacterial peptidoglycan imaging represents a promising strategy for bacterial detection. Peptidoglycan is a network of short peptides that crosslink N-acetylglucosamine (NAG) and N-acetylmuramic acid (NAM) repeat units [23]. Peptidoglycan short peptides contain amino acids in both the l and d configurations, of which the most common d-amino acids are d-Ala and d-Glu. The insertion of fluorescent d-amino acids (FDAAs) can effectively distinguish bacterial and mammalian cells [24]. Bacteria produce new d-amino acids at the fourth or fifth site of peptidoglycan stem peptides by D-amino acid transferase catalysis and l,d- or d,d-peptide transfer reaction [25]. Notably, bacteria can metabolize non-natural d-amino acids into peptidoglycan stem peptides from surrounding mediators via penicillin-binding proteins [26].

Under this mechanism, Kuru et al. synthesized a series of FDAAs (**1**–**4**) in Figure 2 for the first time and successfully labeled and imaged pathogens in living cells [27]. Bacteria hidden in cells are not easily killed by common antibiotics. Hu et al. combined d-alanine with aggregation-induced emission photosensitizers (AIEPSs) and reported that probe **5** can enable fluorescence upon imaging and in situ photodynamic ablation of bacteria in living host cells [28]. Hsu et al. compared FDAA labeling of *Escherichia coli* with a molecular weight of 300–700 Da; they found that FDAAs above 500 Da reduced the pathway to the periplasm and cytoplasm of *E. coli*, and the efficiency of the labeling cell wall was also lower [29]. Feng et al. took advantage of this characteristic to prepare probe **6** by coupling IR-780 iodide with dibenzocycloctyl acetylene. After the bacteria were metabolized by 3-azido-d-alanine, Gram-positive bacteria could be killed directionally under infrared irradiation [30].

### 2.2. Outer-Membrane Glycolipids

The mycomembrane (MM) of *Mycobacteria* and other members of the *Corynebacterium* family is a special outer membrane. The MM is a major defense barrier for *Mycobacterium tuberculosis* and related bacteria, even against antibiotics [31]. As shown in Figure 3, it has the plasma membrane and peptidoglycan layer of bacteria but does not have the teichoic acid of Gram-positive bacteria, nor does it have the LPS of Gram-negative bacteria. In addition, the surface of *Mycobacteria* and Gram-negative bacteria is rich in many glycolipids and lipids, which exert a protective effect, and can only be marked by smaller metabolic marker reagents [32]. Therefore, exotic polysaccharide derivatives are often used to label the surface glycolipid as a portion of Gram-negative and *Mycobacteria*.

The MM comprises a thick arabinogalactan–peptidoglycan polymer covalently linked to an outer lipid layer, which is mainly composed of mycolic acids and organized in an outer-membrane-like structure. Trehalose and arabinogalactan are important raw materials of the MM. The key components of MM glycolipids are trehalose 6,6′-dimycolate (TDM) and trehalose monomethylate (TMM). Trehalose is immobilized on the cell wall of *Mycobacterium* in a non-covalent manner in the form of TMM and TDM. Trehalose is synthesized and esterified in the cytoplasm to form TMM, and TMM is transported to the periplasm. Once in the periplasm, Antige85ABC mycoloyl transferase complex transfers mycolate from TMM to arabinogalactan, forming a network of MM-based covalent mycolates, or from TMM to TDM [33,34,35]. Trehalose exists only in members of the Corynebacterium suborder and is not suitable for typical Gram-negative or Gram-positive bacteria with metabolic markers [32]. Backus et al. combined fluorescein and trehalose to construct probe **7** in Figure 4, which successfully detected *M. tuberculosis* in infected mammalian macrophages [36]. Sahile et al. combined trehalose with 4-N, *N*-dimethylamino-1,8-naphthalimide to form probe **8**, which can be used for no-wash fluorescence imaging and detection of *M. tuberculosis* in sputum samples from tuberculosis-positive patients [37].

The molecular weight, hydrophobicity, and charge of the probe all affect whether the probe can passively diffuse or enter the cell through pore proteins. Therefore, FDAAs are subject to some limitations in detecting Gram-negative bacteria and mycobacteria [38]. 2-keto-3-deoxyoctonicacid (KDO) was found play an important role in the synthesis of membrane LPS in Gram-negative bacteria [39]. Wu et al. incubated azide-modified KDO with Gram-negative bacteria, and probe **9** in Figure 4 was able to accurately ablate Gram-negative bacteria under light [40].

### 2.3. Other Carbohydrates

Bacterial cell-surface polysaccharides are typical drug targets because they play a key role in host colonization, pathogen survival, and immune evasion [41,42]. Analysis of the structure of bacterial polysaccharides revealed the presence of amino-deoxy monosaccharides unique to bacteria and not found in eukaryotes. Highly modified bacterial carbohydrates, such as *N*,*N*′-diacetobacilamine (diNAcBac) and pseudoamino, are common features of bacterial glycans. Bacterial glycans are important molecules mediating host–bacteria interactions [43,44]. Clark et al. designed and synthesized azide-containing analogs of the naturally abundant monosaccharide *N*-acetylglucosamine, FucNAc, bacillosamine, and DATDG (Figure 5). A Staudinger connection between azide polysaccharide and a phosphine probe successfully identified *P. aeruginosa*, *H. influenzae*, *S. aureus*, *Campylobacter jejuni*, and *H. pylori* [45]. Vibhute et al. used the fluorescent dye TamRA-PEG4-DBCO and azide-acetamide-functionalized pseudoamino to conduct a copper-free click reaction to obtain probe **10**, which successfully detected *Bacillus cyuningiensis* and *Bacillus jejuni* [46].

In contrast to mammalian cells, bacteria contain maltodextrin transporters on their surfaces. The unique maltodextrin transport mechanism enables bacteria to absorb maltose, maltotriose, and maltohexose from the outside to maintain glucose content. Zlitni et al. combined maltotriose with Cy7 to construct probe **11**, which is widely used in photoacoustic and fluorescence imaging of bacterial infections [47]. Ning et al. combined maltose hexose with two fluorescent dyes to compose probes **12** and **13** in Figure 6. The probe was a thousand times more specific to bacteria than mammalian cells and could distinguish between live bacteria and inflammation caused by LPS or metabolically inactive bacteria [48].

## 3. Identification of the Cell Wall Components of Pathogenic Bacteria

The classification and identification of bacteria is the first step in the prevention and control of bacteria and in reducing transmission, as well as in reducing the overuse of antibiotics and ensuring effective treatment. Different antibiotics are selected depending on the type of pathogenic bacteria. One of the biggest differences between pathogenic bacteria is the cell wall, which generally differs between Gram-positive and Gram-negative bacteria [24]. The most common binding sites for probes are peptidoglycan and teichoic acid from the cell wall of Gram-positive bacteria and LPS from the cell wall of Gram-negative bacteria.

### 3.1. Peptidoglycan

Peptidoglycans are found in the cell walls of both Gram-positive and Gram-negative bacteria. As shown in Figure 7, peptidoglycan is a multilayer reticular macromolecular structure formed by the polymerization of NAM and NAG with four to five amino acid peptides. Each NAM leads to an oligopeptide chain, which is linked to NAM on the adjacent polysaccharide chain. Two parallel sugar chains are crosslinked to form a network, thus forming a layer of peptidoglycan [49]. Unlike FDAAs, fluorescent probes with antibiotics, lysozymes, and boric acid as recognition receptors usually cannot penetrate the outer membrane of Gram-negative bacteria and bind to peptidoglycan due to their high molecular weight. Therefore, this kind of probe is usually used as a universal fluorescent probe for Gram-positive bacteria.

Vancomycin (VAN) is a glycopeptide antibiotic that can partially bind to d-Ala-d-Ala on the stem peptide of the peptidoglycan subunit to interfere with the synthesis and sterilization of cell walls of Gram-positive bacteria [50]. Mills et al. designed a no-wash probe (**14**) in Figure 8 based on VAN and an environmental merocyanine dye that quickly and specifically detect Gram-positive bacteria in human lungs [51]. Ning et al. obtained probe **15** by coupling active polythiophene derivatives with VAN and α-methoxy-o-amino-polyethylene glycol through an ester–amine reaction and achieved fluorescence imaging of Gram-positive bacteria [52]. Previous reports have shown that the multivalent interaction of VAN dimers or oligomers can considerably enhance their binding potential. Based on this method, Feng et al. synthesized probe **16** for selective identification, naked-eye detection, and photodynamic killing of Gram-positive bacteria, as well as vancomycin-resistant *Enterococcus* [53,54]. Zhong et al. modified VAN on carbon quantum dots (CQDs) through the OH- and COOH-rich functional groups on the surface of CQD-synthesized fluorescent probe CD@Van and identified Gram-positive bacteria in orange juice and other beverages [55]. Teicoplanin (TEIC) is a glycopeptide antibiotic used to treat many Gram-positive bacterial infections. Whereas TEIC is not routinely used in hospitals, it is effective at killing *S. aureus* (including methicillin-resistant *S. aureus*) with fewer side effects than VAN [56]. Wang et al. successfully used TEIC synthetic probes covalently connected with alkaline earth sulfide nanoparticles for Gram-positive in vivo and in situ imaging [57].

Lysozyme, also known as 1,4-*N*-acetylenic enzyme, hydrolyzes the 1,4-glycoside bond between NAM and NAG [58]. Arabski et al. modified lysozyme with carboxyl fluorescein, a water-soluble triazine coupling agent, achieving detection of the distribution and number of Gram-positive bacteria in the environment [59]. Zheng et al. developed a fluorescein isothiocyanate-labeled lysozyme probe, which can be used not only as a platform for pathogen detection but also as a tracer reagent for microbial populations in antibacterial tests [60]. Wheat germ agglutinin (WGA) specifically binds to the glucose residues of NAG [61]. Due to the blocking of the outer membrane of Gram-negative bacteria, WGA can only bind Gram-positive bacteria. Zhang et al. combined WGA with fluorescein isothiocyanate for directional staining of Gram-positive bacteria [62]. Prolonged staining can also bind the probe to Gram-negative bacteria [63].

### 3.2. Teichoic Acid

Teichoic acid is a unique component of the cell wall of Gram-positive bacteria. It is a weakly acidic substance made of ribitol or glycerol residues linked to each other by phosphate groups [64]. Teichoic acid is divided into wall teichoic acid and lipoteichoic acid (LTA). Wall teichoic acid is linked to NAM residue of peptidoglycan by phosphodiester bonds and does not contact the plasma membrane. LTA passes through the polypeptide polysaccharide layer and covalently connects the oligosaccharide portion of glycolipids in the plasma membrane with terminal phosphoric acid. Naik et al. synthesized probes **17** and **18** in Figure 9 using sulfonate groups and tetraphenyl ethylene (TPE) based on the NH_3_^+^ group on the backbone of the teichoic acid as recognition objects. These probes were not only used in Gram-positive bacteria imaging but also in apple juice contaminated by *S. aureus* as an example [65]. Hu et al. found that aggregation-induced emission (AIE)-active 2-{[(diphenylmethylene)hydrazono]methyl}phenol (DPAS) derivatives can selectively enrich lipid droplets [66,67]. Therefore, probe **19** in Figure 9 containing morpholine and naphthalene units was designed and synthesized; via a strong hydrophobic effect, DPAN groups easily insert Gram-positive bacteria and fungi of the thick and loose outer layer. However, the alkalinity of morpholine moiety in probe **19** is compatible with the acid structures embedded in the peptidoglycan layer of Gram-positive bacteria, although it is not suitable for binding with the outer-layer structure of fungus, causing the probe to easily fall of. Moreover, LTA was reported to bind with probe **19**, benefiting the anchoring of probe **19** on the cell wall of Gram-positive bacteria. For Gram-negative bacteria, the outer membrane inhibits the insertion of probe **19** moieties [68].

### 3.3. Lipopolysaccharide

LPS is a unique component of the cell wall of Gram-negative bacteria [69]. Composed of core polysaccharides, *O*-polysaccharide side chains, and lipid A, LPS comprises endotoxins and group-specific antigens. Usually, the molecular weight of LPS is greater than 10,000 Da, with a complex structure, which varies among groups and even strains [70]. Therefore, antibiotics or antimicrobial peptides are often used to recognition original of LPS. Polymyxin is a polypeptide antibiotic, which inhibit and kill Gram-negative bacteria by electrostatic and hydrophobic combination with LPS and membrane phospholipid of the bacterial outer membrane [71]. Bao et al. synthesized probe **20** by combining polymyxin B with a TPE derivative, which can selectively detect photodynamic inactivated Gram-negative bacteria [72]. Due to the presence of polyamine groups in colistin E, Ryu et al. synthesized probes **21**, **22**, and **23** in Figure 10 of colistin E and Cy3 for rapid clinical screening and stratification of Gram-negative bacterial infections [73]. Positively charged pyridine salts can effectively accumulate with negatively charged LPS by electrostatic forces. Liu et al. designed AIE probe **24** in Figure 10 with an ethyl chain and a pyridine salt with a positive charge. The ethyl chain and positively charged pyridine salts allow probe **24** to be easily inserted into the cell membrane for binding to LPS. Probe **24** can not only image *E. coli* but also produce reactive oxygen species to kill it when exposed to light at 530 nm [74].

### 3.4. Carbohydrate Compound

In addition to peptidoglycan and LPS, the bacterial surface is rich in many glycoproteins, which are mainly composed of sugar. Boric acid is a well-known chemical unit that can bind with sugars, so the molecules containing boric acid are gradually expanded from the detection of various sugars to the detection of pathogenic bacteria [75]. Tsuchido et al. first used a boric acid probe to detect bacteria and modified the fourth generation of poly(amidoamine) with boric acid (Figure 11). The aggregation of probes and various bacteria was pH-dependent. Both Gram-positive and Gram-negative bacteria formed aggregates at alkaline pH values, whereas only Gram-positive bacteria formed aggregates at neutral pH values [76].

Hu et al. designed a boric acid amphiphilic fluorescent probe (**25**) with AIE characteristics, which can strongly interact with LPS through a variety of interactions, including hydrophobicity, electrostatic interaction, and esterification of boric acid groups with 1,2- and 1,2-diol units in disaccharides, and be used to identify Gram-negative bacteria [77]. Bacteria cannot be identified by a single boric acid element. Kwon et al. synthesized probe **26**, which was generally selective for Gram-positive bacteria, by constantly modifying the head and tail groups of the probe with BODIPY as the fluorescence group and boric acid as the recognition group. Probe **26** in Figure 12 has the best selectivity for Gram-positive bacteria and has the least staining process for sample detection and enrichment of live bacteria. Probe **26** can quickly and selectively detect Gram-positive bacterial infections. It can be used to evaluate the status of Gram-positive bacteria in sewage sludge flora, and it can also be used for in vivo fluorescence imaging of keratitis mice [63].

### 3.5. Other Components

Antimicrobial peptides act on the membrane, forming ion channels through the membrane, destroying its integrity, and causing leakage of cell contents, thus killing bacteria. However, the mechanism by which antimicrobial peptides kill bacteria has not been fully elucidated [79]. As the recognition element of the fluorescent probe, the selection of an affinity object of the antimicrobial peptide determines the object of probe application. Liu et al. synthesized probe **27** in Figure 12 by coupling an LPS-conjugated antibacterial agent (YVLWKRKRKFCFI-amide) with photosensitizer protoporphyrin IX to inactivate a fluorescent imaging signal and photodynamics of Gram-negative bacteria [78]. Marc Vendrell et al. developed a new fluorescent amino acid, Phe-BODIPY, with excellent photophysical properties (Figure 13). Subsequently, after identifying antifungal peptides with affinity for Candida cells, the team further synthesized fluorescent linear peptide **17** and used it for rapid detection of Candida in human urine samples [80]. Significantly, the minimal cell permeability and stability of antimicrobial peptides in biological environments limit their widespread use [81]. Jiao et al. synthesized a hepta-dicyanomethylene-4*H*-pyran appended *β*-cyclodextrin (DCM_7_-*β*-Cd), which can form supramolecular-peptide-dots (Spds) with 1-bromonaphthalene-modified polypeptides (Figure 14). The fluorescent protein Spd-4 was prepared, which penetrated the cell wall into *E. coli* and *S. aureus* more easily than the peptide alone (P4-FITC) [82].

Aptamers are oligonucleotide sequences with high specificity and affinity for target substances screened from random oligonucleotide libraries based on SELEX technology. Therefore, aptamers with a high affinity for specific bacteria are often selected as recognition elements to construct fluorescent probes [83]. Hu et al. fabricated a fluorescent probe (CQDs-MNPs) for *E. coli* based on aptamer-labeled CQDs and complementary DNA-labeled magnetic nanoparticles. The CQDs-MNP method was used to analyze milk samples contaminated with *E. coli*, and the results were consistent with those obtained via the plate-counting method [84]. Yang et al. constructed fluorescent probes by combining fluorescent nanoparticle AgNCs with aptamers that specifically recognize *S. aureus*. The probe successfully detected in naked eyes and killed *S. aureus*. As shown in Figure 15, the aptamer of *S. aureus* affinity was linked to a G-rich sequence, and fluorescence was enhanced when *S. aureus* was captured by PLA-AgNCs. This method shows considerable application value in biomedical engineering and food packaging [85].

## 4. Identification of Endogenous Enzymes of Pathogenic Bacteria

Enzymes are involved in many catalytic processes, such as metabolism, nutrition, and energy transformation, and are closely related to life processes. Caspase-1 induces programmed cell death in macrophages infected with bacteria [86]. Qi et al. developed a new molecular probe with two moieties, an AIE molecule and an enzyme-responsive peptide linker (NEAYVHDAP). Residues generated by caspase-1-specific cleavage probes self-assembled into aggregates. These aggregates not only produced bright fluorescence under 405 nm light but also produced reactive oxygen species to kill bacteria [87]. In addition to the characteristic enzymes produced by the target of bacterial infection, bacteria themselves have characteristic endogenous enzymes [88]. For example, NTR, a flavin-containing enzyme, mainly exists in bacteria, archaea, and eukaryotes; alkaline phosphatase, which is highly expressed by bacteria, is mainly located in the periplasmic region of the bacterial surface wall and is associated with the outer membrane; *β*-lactamase, a hydrolytic enzyme, is the main means of resistance to *β*-lactamase antibiotics by drug-resistant bacteria [89,90,91]. Therefore, it is possible to target NTR, alkaline phosphatase, and *β*-lactamase for fluorescence imaging of specific bacteria.

### 4.1. Nitroreductase

NTR produced by bacteria is a kind of flavonoid that uses reduced nicotinamide adenine dinucleotide or reduced nicotinamide adenine dinucleotide phosphate as reduction equivalents and can selectively catalyze the reduction of nitro groups of various aromatic nitro compounds to hydroxylamines or amines to prevent nitro invasion in the environment [92,93]. NTR currently represents an important marker for the detection of a variety of bacteria in clinical diagnosis, as well as food and environmental sample safety assessment. Wangngae et al. synthesized a chalcone fluorescent probe (**28**) in Figure 16 for bacterial detection [94]. Yoon et al. synthesized a resorcinol-based fluorescent probe (**29**) for bacterial detection and rinse-free imaging [95]. Wu et al. synthesized a near-infrared probe (**30**) to specifically detect NTR, which can not only image living bacterial cells but also distinguish cancer from a bacterial infection in a mouse model [96].

### 4.2. Alkaline Phosphatase

Among bacteria-based biomarkers, alkaline phosphatase is an enzyme responsible for the hydrolysis of phosphate esters to promote the release of inorganic phosphorus, which is essential for bacterial cell growth [97]. Zhang et al. designed probe **31** in Figure 17 for specific recognition of alkaline phosphatase by coupling with self-assembled peptides, which performed well in detecting the activity of alkaline phosphatase and in situ imaging of bacteria against *E. coli* [98]. Gwynne et al. designed a colorimetric fluorescent probe (**32**) to detect phosphatases in bacteria. However, probe **32** did not react with *E. faecium*, *P. aeruginosa*, or *E. coli*—only with *S. aureus* [99].

### 4.3. β-Lactamase

*β*-lactam antibiotics contain a *β*-lactam ring made up of four atoms. The four-membered ring of *β*-lactam is the pharmacophore necessary for the antibiotic to function. With the widespread use of these antibiotics, bacteria not only reduce their efficacy by changing the permeability of their cell wall or cell membrane but also produce *β*-lactamase to destroy antibiotics, causing pathogens to develop serious resistance to antibiotics. *β*-lactamase can destroy the *β*-lactam ring in antibiotics, rendering them useless [8,100]. Therefore, targeting *β*-lactamase is of considerable interest for fluorescence imaging of super bacteria. Ma et al. synthesized a near-infrared fluorescent probe (**33**) in Figure 18, by adding cephalosporin to the semi-cyanine skeleton. When *β*-lactamase produced by bacteria reacted with the probe, the four-membered ring was opened, and part of the cephalosporin released semi-cyanine, resulting in a sharp increase in fluorescence intensity [101]. As a *β*-lactamase, AmpC often appears in *Enterobacteriaceae* and *Pesudomonas pyocyaneum*, which normally produce a small amount of AmpC; however, in the presence of *β*-lactam antibiotics, they express a considerable amount of AmpC [8,91]. Xing et al. prepared a probe (**34**) in Figure 18 for selective detection of the AmpC enzyme. A large sterically hindered methoxy sulfhydryl group was introduced to the probe at position **7** of cephalosporin, enabling it to be targeted and recognized by AmpC enzymes. The conformation of the probe was changed by AmpC, which led to the phenomenon of AIE [102]. Mehta et al. conjugated an environmentally sensitive fluorophore onto a mercaptan-based scaffold of New Delhi metallo-*β*-lactamase (NDM) inhibitors (probe **35**) to monitor the dynamic metalation of NDM in *E. coli* and for fluorescence imaging [103].

## 5. Nonspecific Sites Identify Pathogens

Phospholipids, phosphatidylglycerol, and cardiolipin in bacterial cell membranes provide bacteria a negatively charged surface. The abundance of glycoproteins on the bacterial surface makes it negatively charged, even in neutral or weakly alkaline environments [104]. Therefore, below, we summarize the current research progress of organic small-molecule probes, metal complex probes, and conjugated polymer probes for identification and imaging of bacteria by electrostatic and hydrophobic interactions.

### 5.1. Organic Small-Molecule Probe

In recent years, organic small-molecule fluorescent probes have played an increasingly important role in pathogen infection detection and imaging of living systems due to their advantages of simple structure, convenient synthesis, high sensitivity, high selectivity, and strong anti-interference ability [105]. Usually, organic cations containing quaternary ammonium, phosphorus, pyridine, and imidazole are coupled with fluorescent dyes to bind negative charges on the surface of bacteria. Li et al. designed an organic salt photosensitizer with aggregation-induced luminescence properties (probe **36** in Figure 19). The probe interacts with the bacterial surface through electrostatic attraction and hydrophobic effects to kill bacteria and cancer cells without affecting normal cells [106]. Lee et al., for the first time, used a water-soluble near-infrared probe (**37**) with aggregation-induced luminescence, which was simultaneously used for the identification of 3-second washing-free Gram-positive bacteria and the dual application of photodynamic antibiotics [107]. Shi et al. synthesized probes **38** and **39** in Figure 19 by introducing one or two positive charges on the pyridine side of triphenylmethane derivatives and found that these two novel red AIEPSs can image bacteria and destroy Gram-negative bacteria [108]. Shi et al. designed and synthesized a series of imidazole-type ionic liquids based on AIE. By changing the carbon chain length of substitution at the N3 position of the imidazolium cation, synthetic probes **40**, **41**, and **42** were successfully used to image bacteria in cells, demonstrating that the longer the substituted chain, the more obvious the killing effect on bacteria [109]. Panigrahi et al. synthesized a positively charged probe (**43**) through a one-step Schiff base condensation reaction. Effective destruction of membrane integrity by Gram-positive and Gram-negative bacteria interacting with the probe aggregate was also observed through strong electrostatic binding to the bacterial surface and lighting of the bacteria [110]. Long et al. developed an imidazole-derived pyrene aggregate (probe **44**) which breaks down and opens with a significant increase in fluorescence intensity when the nonemissive aggregates bind to the surface of anionic bacteria. In addition, 10 bacteria and 14 clinically isolated multidrug-resistant bacteria were rapidly identified, and their Gram-staining characteristics were determined based on change analysis of fluorescence emission spectral data [111].

### 5.2. Metal Complex Probe

Metal complexes possess many excellent photophysical properties, such as long luminescence lifetime (100 ns~1 ms), high Stokes displacement, enhanced photostability, high quantum yield, high signal-to-noise ratio, easy tuning of luminescence wavelength, and simple synthetic route. In recent years, Ir(III), Ru(II), Os(II), Re(I), Pt(II), Au(I), Cu(I), and Zn(II) complexes have been used in the field of bacterial detection and imaging [112]. The luminescence properties of rare earth complexes are similar to those of AIEgens, as rare earth complexes generally have high luminescence efficiency both in aggregate and solid form [113]. Gupta et al. developed a new ring metal iridium (III) complex (**1**–**3**) (Figure 20), a cationic Ir(III) complex combined with negatively charged phosphoric groups on LPS and LTA through electrostatic interaction to form LPS/LTA-Ir(III) aggregates, resulting in strong intermolecular p–p stacking to achieve rapid detection and suppression of drug-resistant bacteria [114]. Zhu et al. reported a luminescent probe (**45**) in Figure 21 based on the formation of 2,6-bis(benzimidazol-2′-yl) pyridine with hexaethylene glycol methyl ether ([Pt(N^^^N^^^N)Cl]^+^) chloroplatinic (II) complex. The probe binds negatively charged LPS to form LPS-Pt (II) aggregates and quickly distinguish Gram-negative and Gram-positive bacteria within 5 min without washing [115]. Leevy et al. synthesized probes **46** and **47** and, for the first time, used zinc (II) dipyridinium methylamine to coordinate with anionic phospholipids on the outer surface of pathogen membrane through a zinc (II) metal center for fluorescence imaging of *E. coli*, *P. aeruginosa*, and *S. aureus* [116]. Under this influence, Cabral et al. synthesized probes **48** and **49** in Figure 21 using zinc (II) chelate as a recognition group to detect bacteria in 2–3 s without specificity or generality [117].

### 5.3. Conjugated Polymer Probe

Conjugated polymers have multiple π-conjugated continuous repeating units, and each unit cooperates to achieve effective electron coupling. Therefore, the energy captured by conjugated polymers can migrate freely, along with the long-range electron skeleton, achieving efficient energy transfer, which encodes polymers with excellent light-capture performance and signal amplification effects. Conjugated polymers have been widely used in chemical and biological sensing, cell fluorescence imaging, diagnosis and treatment of pathogen infection, and other fields. Zhou et al. synthetically conjugated an oligoelectrolyte probe (**50**) in Figure 21 to specifically identify Gram-positive bacteria. Although probe **50** carries a large positive charge, the presence of an outer membrane on the surface of Gram-negative bacteria makes it difficult to insert [118]. Bai et al. synthesized a cationic polyfluorene derivative probe, PFP−NMe_3_^+^/CB[7] (Figure 22), which could form a supramolecular complex with cucurbit[7]uril (CB[7]) that could be reversibly decomposed by amantadine (AD) to form a more stable CB[7]/AD complex and release the probe. PFP is an amphiphilic structure that binds to negatively charged pathogen membranes through polyvalent interactions. Because CB[7] can bury the side-chain alkyl group and reduce the hydrophobic interaction of the probe on the surface of the pathogen, the probe exhibits a different interaction pattern with the pathogen before and after assembly with CB[7]. A variety of pathogens were successfully identified by changes in optical signals before and after assembly [119]. In subsequent studies, the team also synthesized a cationic polythiophene derivative, which assisted in standard linear discriminant analysis based on changes in polymer fluorescence intensity to achieve rapid and simple identification of viruses and microorganisms [120]. With the development of organic luminescence materials, the development of AIE polymer has attracted increasing attention [121]. Zhang et al. synthesized a cationic AIE polycarbonate probe (**51**) with excellent water solubility, showed excellent imaging and antibacterial ability on *E. coli.* and *S. aureus*. Compared with probe **51**, a mixed-charge AIE polycarbonates probe (**52**) allowed for the rapid and selective imaging of *S. aureus* but not *E. coli* [122].

## 6. Conclusions

In this tutorial review, we summarized the recent research progress with respect to construction strategies for bacterial detection using fluorescence probes. We first discussed the strategy of labeling the surface components of bacteria with small molecular organic fluorescence probes in a direct or indirect manner. Although those metabolic fluorescent molecules are highly selective, they are limited by bacterial metabolism. Moreover, they showed some limitations in the detection of bacteria in complex environments. Next, we outlined the fluorescent probes constructed through the binding of specific components on the surface of bacteria, such as the binding of glycopeptide antibiotics in the cell wall and the affinity of boric acid derivatives to carbohydrates. We also mentioned the detection strategy involving the catalytic reaction of specific groups by bacterial endogenous enzymes, such as nitrate reductase, alkaline phosphatase, and *β*-lactamase. This kind of method has certain advantages for the detection of super bacteria. In addition, the electrostatic and hydrophobic interaction between the probe and the bacterial surface is currently a hot spot in bacterial detection research. According to the conformation of the probe, this method can distinguish Gram-negative bacteria from Gram-positive bacteria and even roughly judge the types of bacteria according to the ratio of fluorescence signals before and after binding to bacteria. However, the high detection limit of this kind of probe is still subject to some limitations in the field of bacterial detection. At present, the detection requirement of most pathogens in various countries is zero detection, and the detection range of fluorescent probes that only rely on hydrophobic and electrostatic interaction as identifying elements is usually between 10^4^ and 10^8^ CFU/mL. Moreover, the probe is easily affected by cations and surfactants in the environment to be measured. Although the use of highly specific recognition elements, such as antimicrobial peptides and aptamers, to construct probes can effectively detect the presence of pathogens, they are difficult to synthesize and easily degraded, so they cannot be widely used.

A recently proposed strategy based on fluorescent probes has attracted considerable interest for the detection of bacteria due to its high selectivity, fast response, and simple operation. We hope that this review can stimulate this research direction in bacterial detection and imaging in the future.

## Figures and Tables

**Figure 1 molecules-27-06440-f001:**
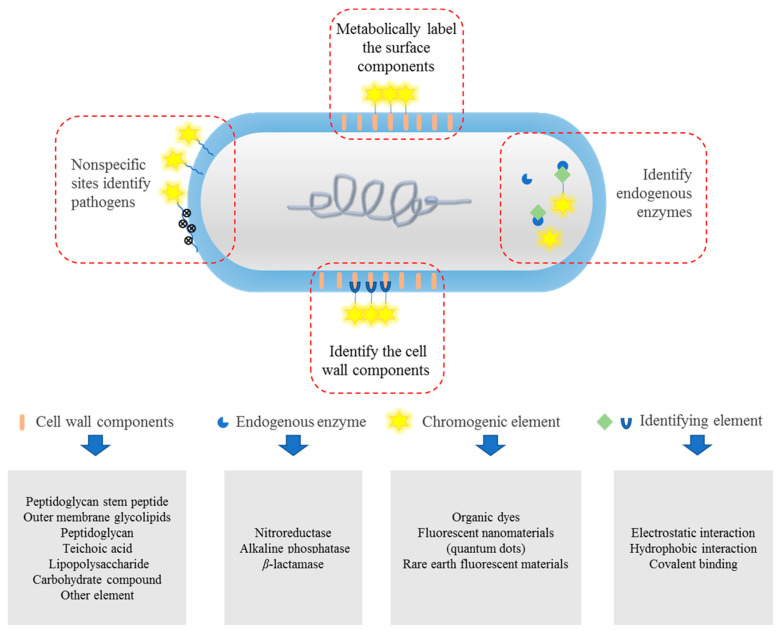
Pathogenic bacterial identification strategies using fluorescent probes.

**Figure 2 molecules-27-06440-f002:**
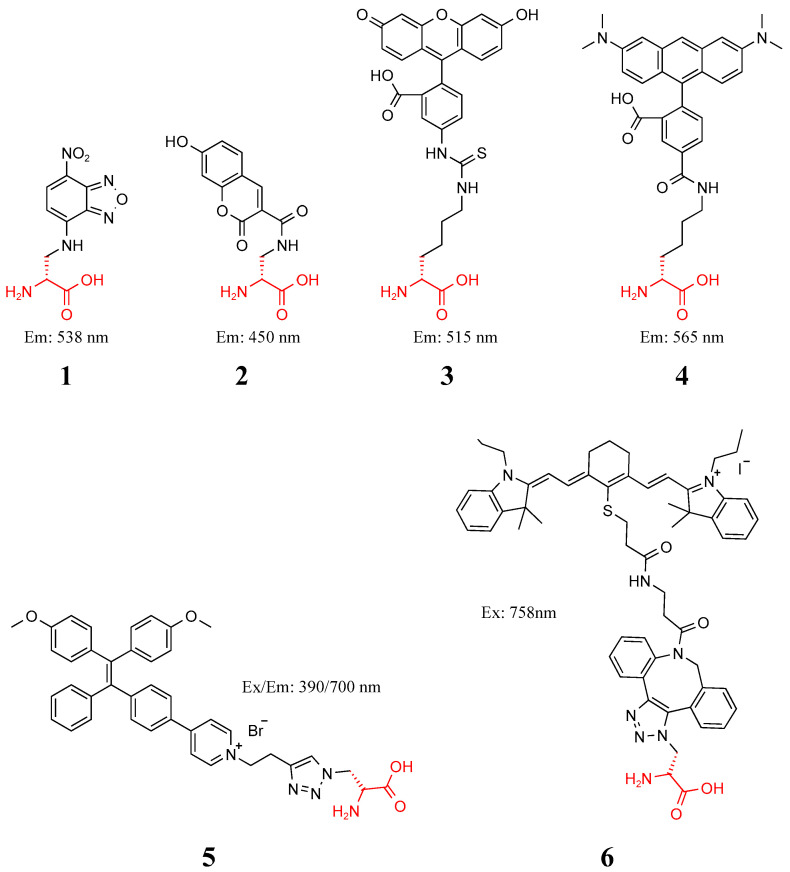
Structure of probes **1**–**6**.

**Figure 3 molecules-27-06440-f003:**
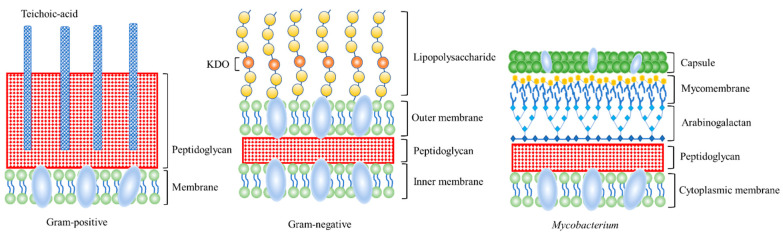
Cell wall structure of Gram-positive bacteria, Gram-negative bacteria, and *Mycobacterium*.

**Figure 4 molecules-27-06440-f004:**
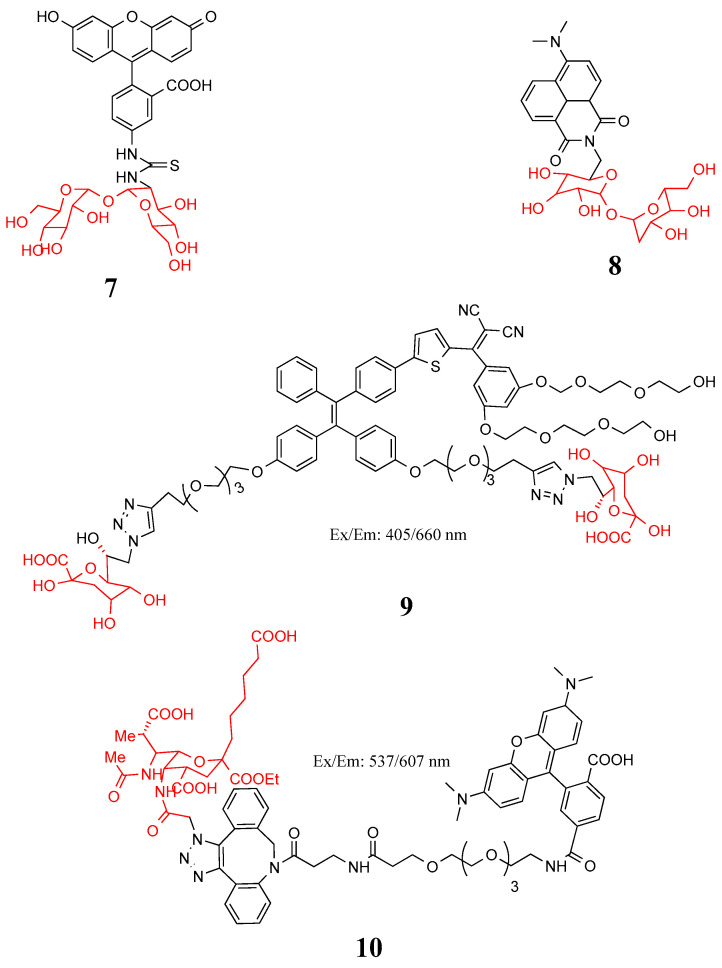
Structure of probes **7**–**10**.

**Figure 5 molecules-27-06440-f005:**
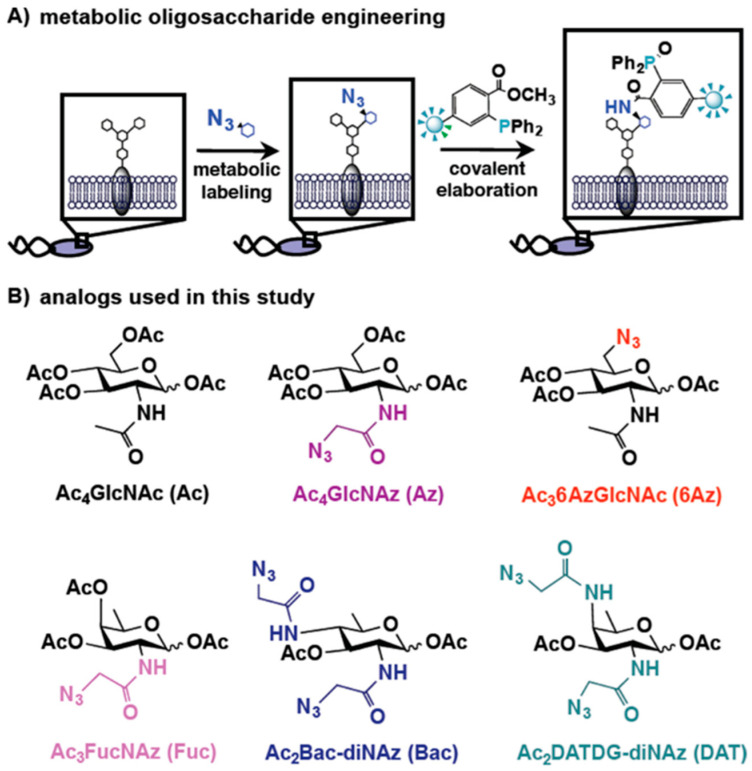
Schematic overview of metabolic oligosaccharide engineering (MOE) experiments and structures of monosaccharide analogs used in this study. (**A**) In MOE experiments, cells are first metabolically labeled with an unnatural azide-containing sugar. Azide-bearing glycans then undergo Staudinger ligation with phosphine probes to enable their detection on cells or in cell lysates. (**B**) Metabolic incorporation of azide-containing analogs of the naturally abundant monosaccharide GlcNAc, as well as of the rare bacterial monosaccharides FucNAc, bacillosamine, and DATDG, was explored in this work. Reproduced with permission from Ref. [45]. Copyright 2016, American Chemical Society.

**Figure 6 molecules-27-06440-f006:**
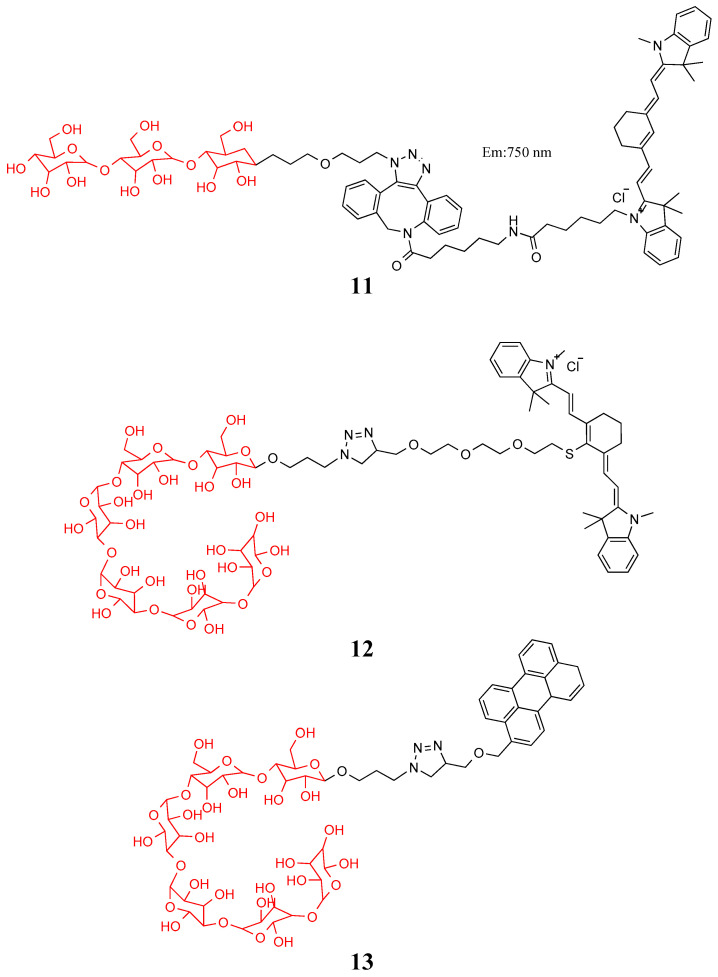
Structure of probes **11**–**13**.

**Figure 7 molecules-27-06440-f007:**
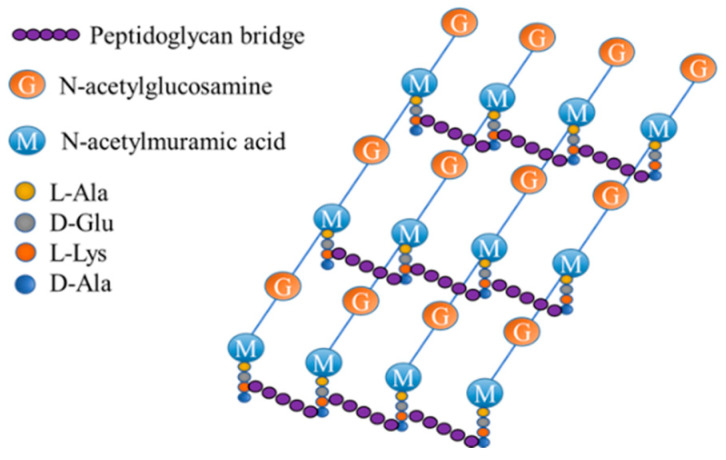
Structure of peptidoglycan from *S. aureus*.

**Figure 8 molecules-27-06440-f008:**
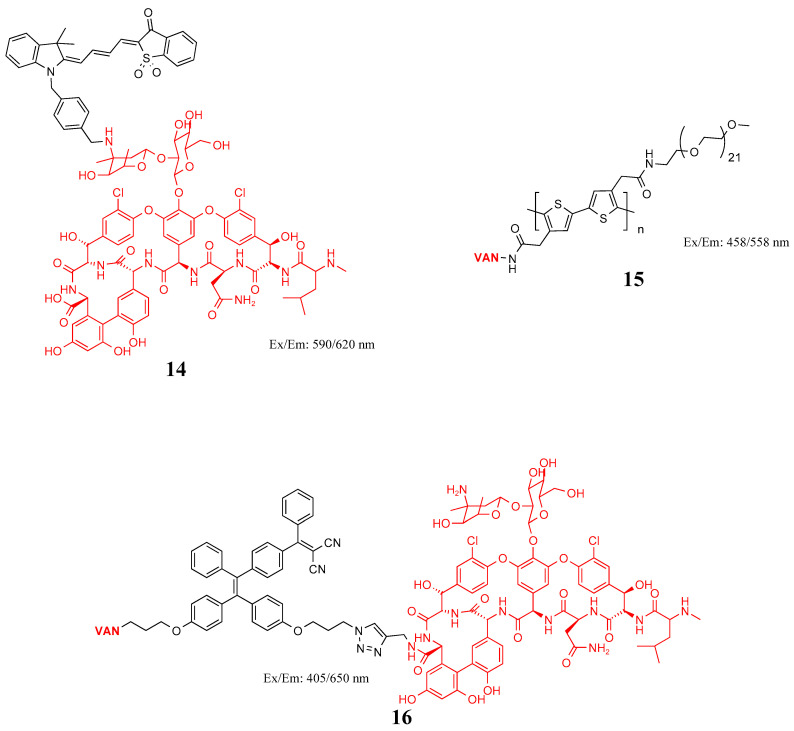
Structure of probes **14**–**16**.

**Figure 9 molecules-27-06440-f009:**
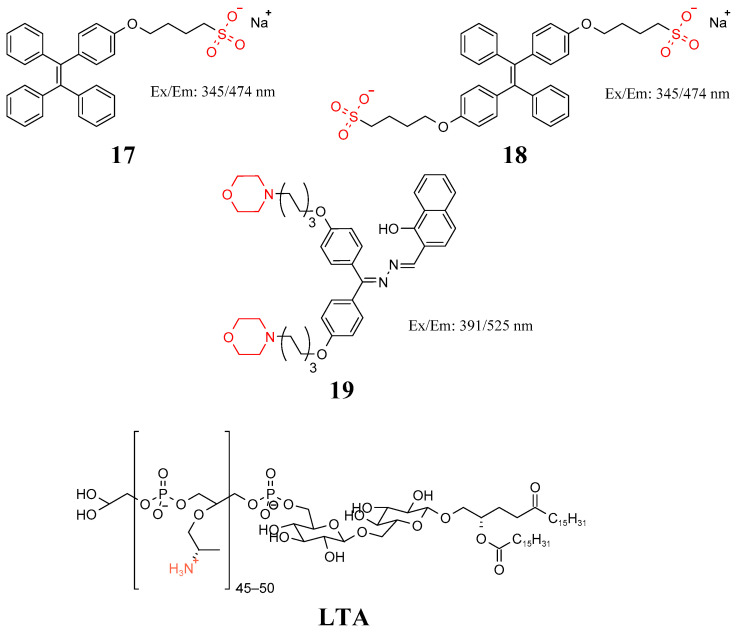
Structure of probes **17**–**19** and LTA.

**Figure 10 molecules-27-06440-f010:**
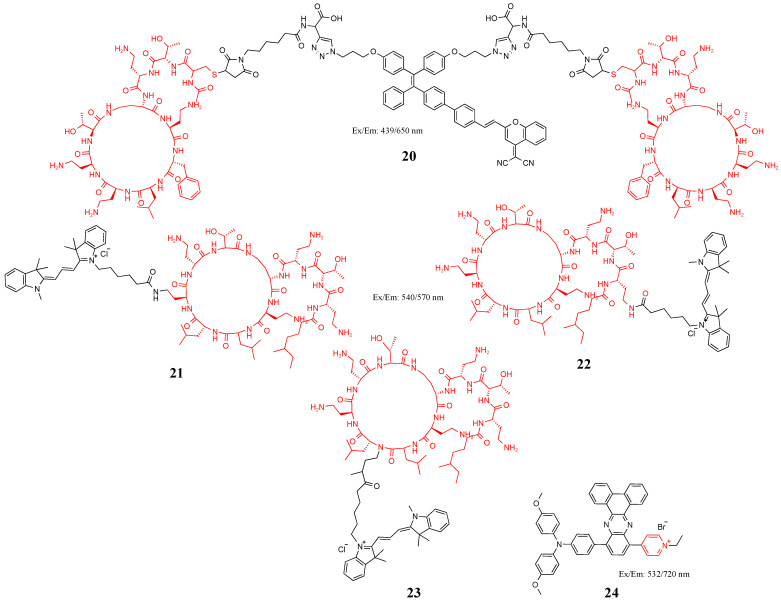
Structure of probes **20**–**24**.

**Figure 11 molecules-27-06440-f011:**
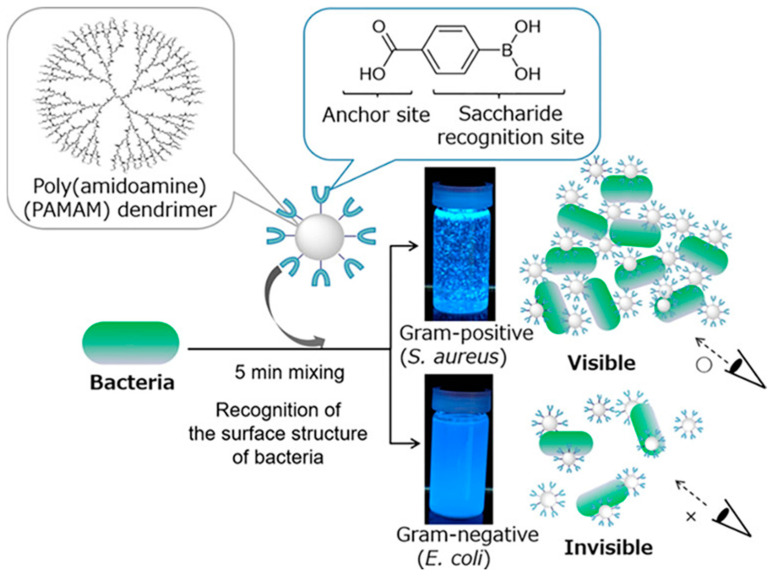
Schematic complex of a boronic-acid-modified poly(amidoamine) dendrimer structure and application. Reproduced with permission from Ref. [76]. Copyright 2019, American Chemical Society.

**Figure 12 molecules-27-06440-f012:**
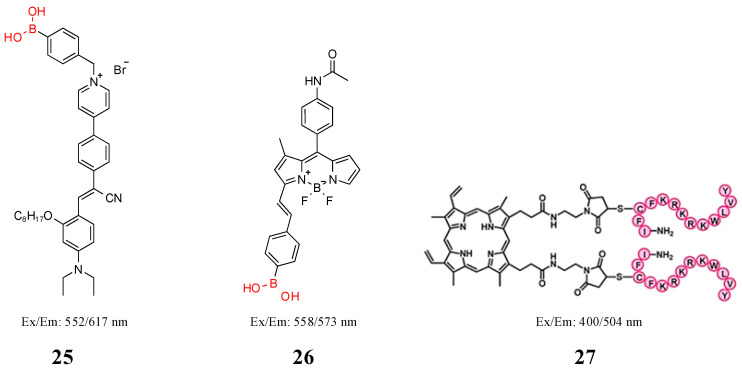
Structure of probes **25**–**27**. Reproduced with permission from Ref. [78]. Copyright 2012, American Chemical Society.

**Figure 13 molecules-27-06440-f013:**
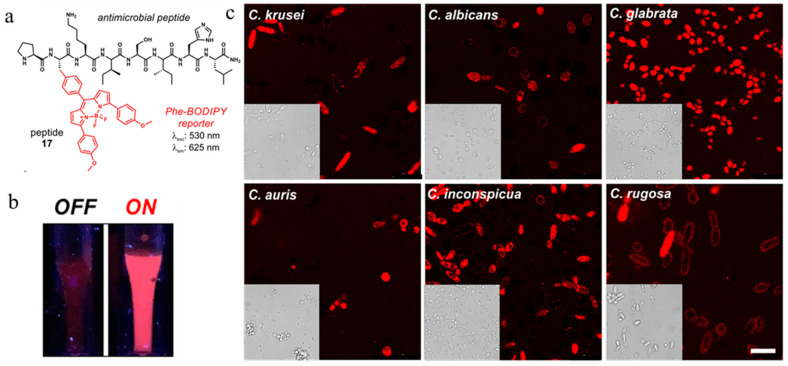
(**a**) Chemical structure of peptide **17**. (**b**) Photos of solutions of probe **17** (30 μM) under excitation with a 365 nm UV-lamp in PBS (**left**) and liposomes (**right**). (**c**) Fluorescence live-cell confocal microscopy of different Candida strains after 1 h incubation with peptide **17** (10 μM) in PBS at 37 °C without any washings. λexc: 575 nm, λem: 600–650 nm. Scale bar: 10 μm. Reproduced with permission from Ref. [80]. Copyright 2022, Wiley-VCH GmbH.

**Figure 14 molecules-27-06440-f014:**
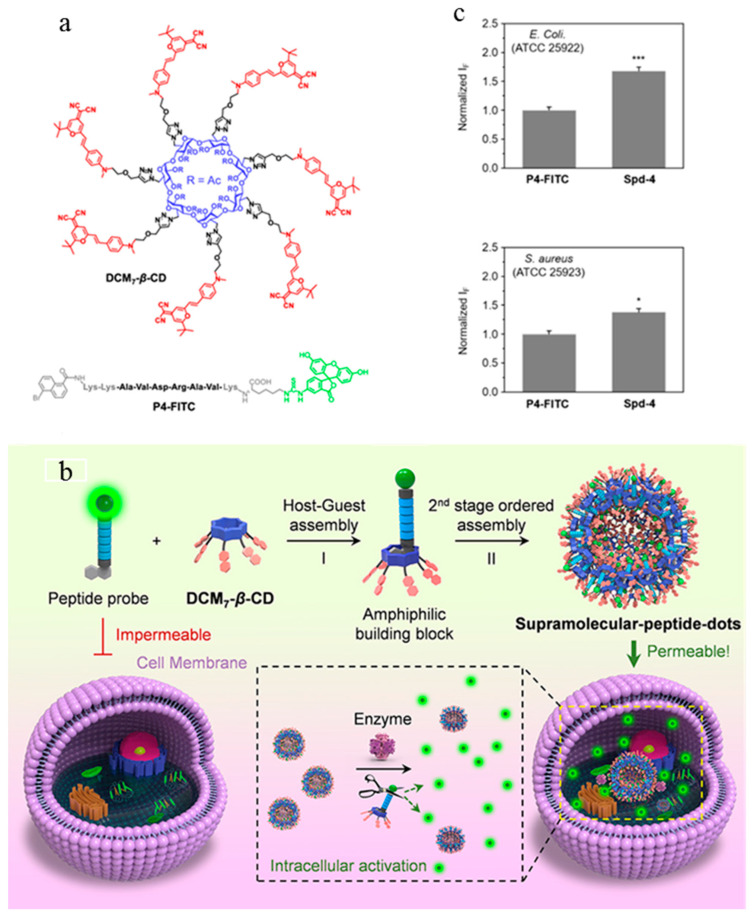
(**a**) Structure of the DCM_7_-*β*-CD and fluorescent antimicrobial (P4-FITC) peptide. (**b**) Schematic illustration of the sequential host–guest and second-stage ordered self-assembly between probes and DCM_7_-*β*-CD to provide so-called Spds with enhanced cellular uptake and functional imaging features. The fluorescent green dots indicate individual FITC moieties. (**c**) Fluorescence quantification of *E. Coli*. and *S. aureus* treated with P4-FITC (1 μM) or Spd-4 (P4-FITC/DCM_7_-*β*-CD = 1 μM/1 μM). * *p* < 0.05, *** *p* < 0.001. Reproduced with permission from Ref. [82]. Copyright 2020, American Chemical Society.

**Figure 15 molecules-27-06440-f015:**
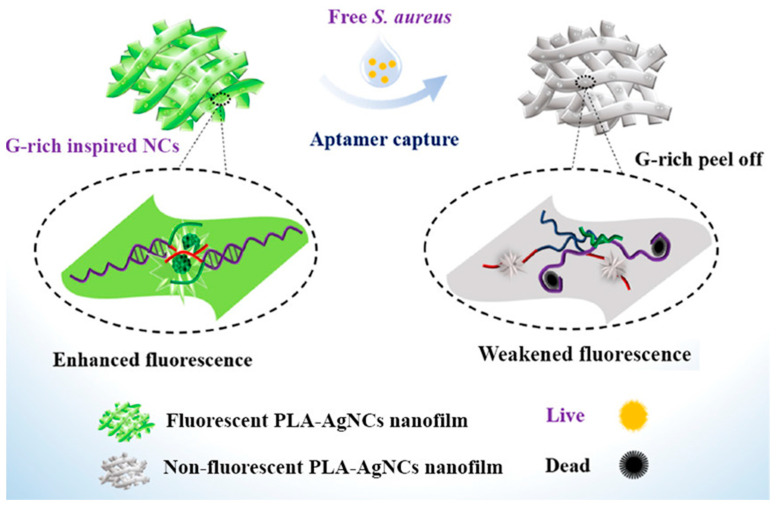
Mechanism of aptamer-enhanced fluorescence and antibacterial activity of DNA-AgNCs in electrospinning film. Reproduced with permission from Ref. [85]. Copyright 2021, American Chemical Society.

**Figure 16 molecules-27-06440-f016:**
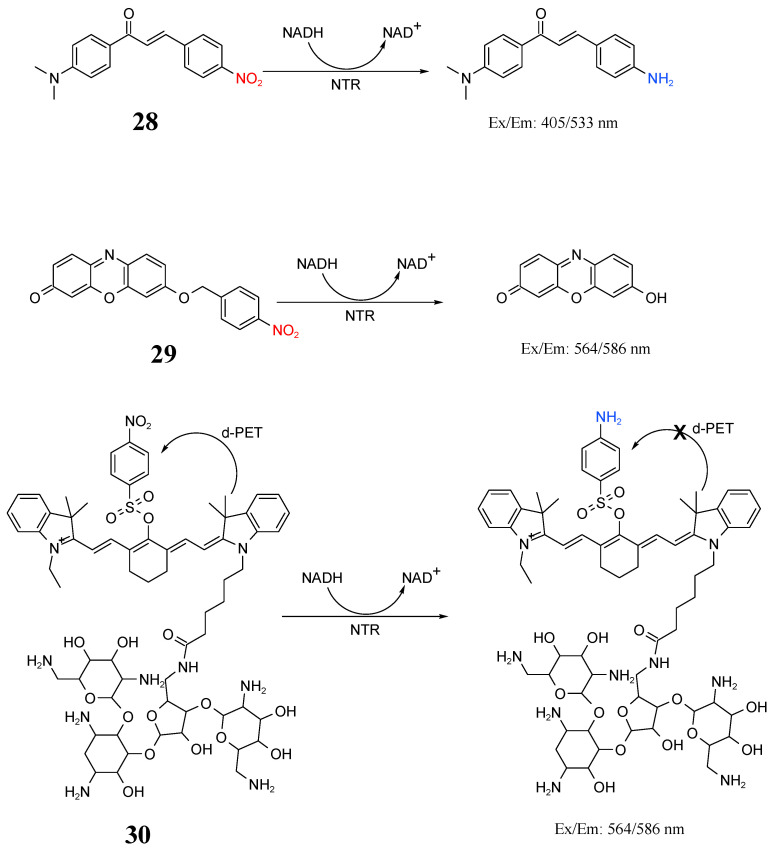
Structure and identification mechanisms of probes **28**–**30**.

**Figure 17 molecules-27-06440-f017:**
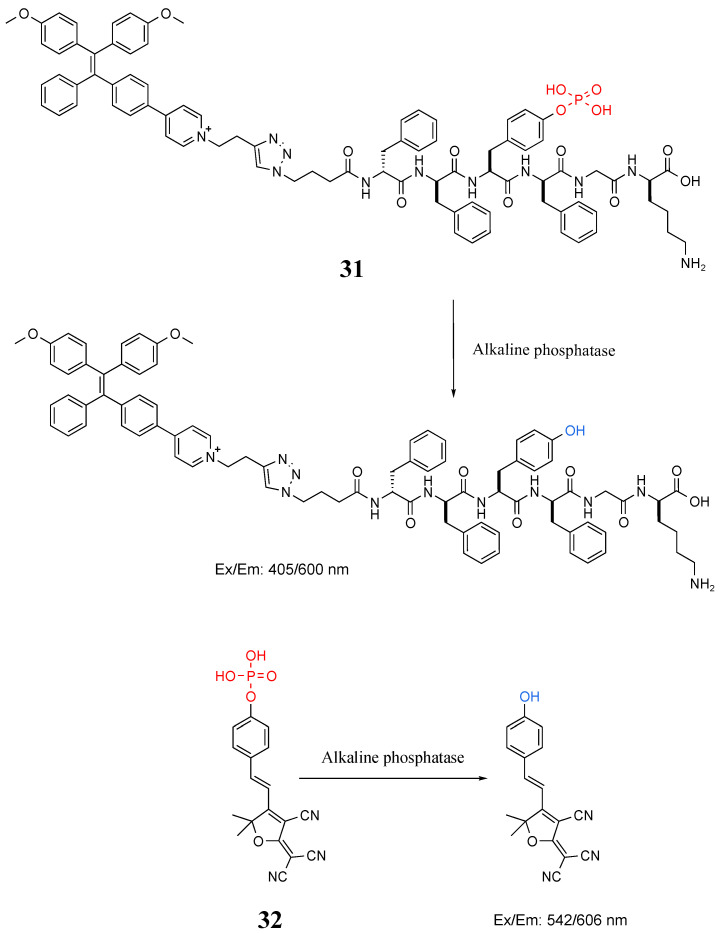
Structure and identification mechanisms of probes **31** and **32**.

**Figure 18 molecules-27-06440-f018:**
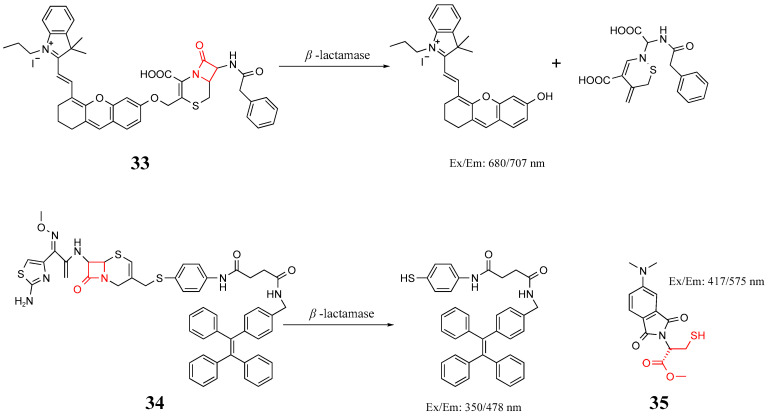
Structure and identification mechanisms of probes **33**–**35**.

**Figure 19 molecules-27-06440-f019:**
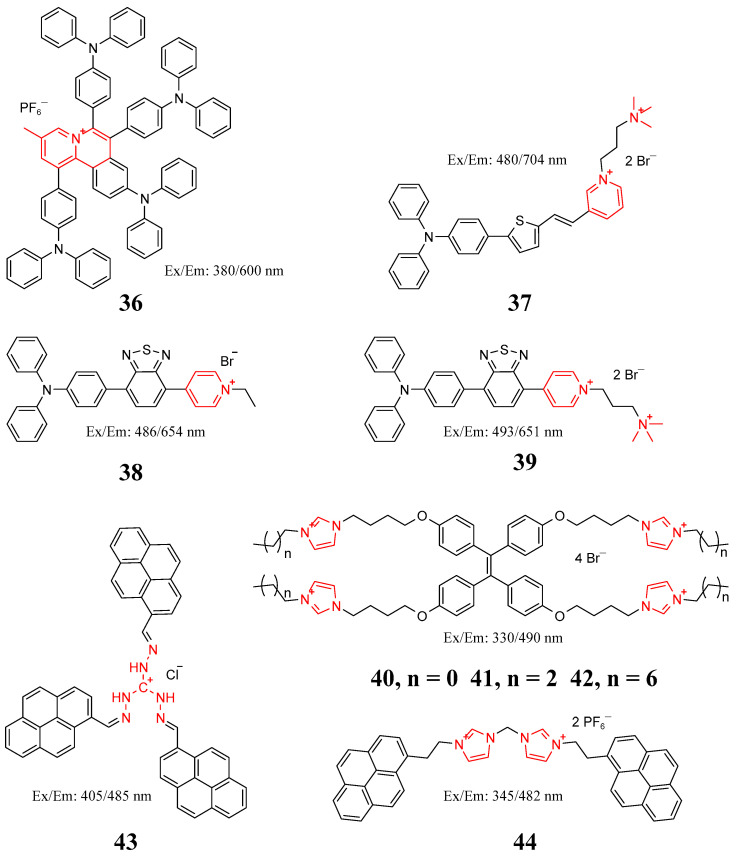
Structure of probes **36**–**44**.

**Figure 20 molecules-27-06440-f020:**
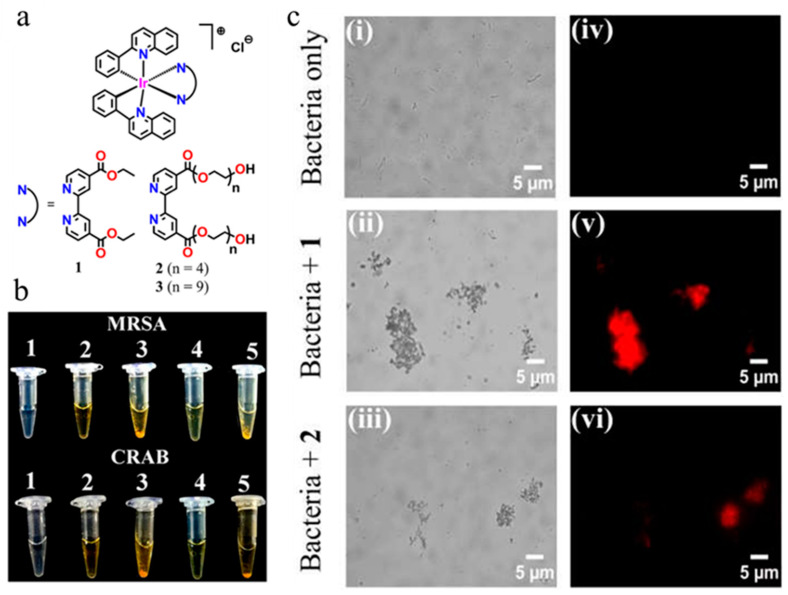
(**a**) Structures of cyclometalated Ir(III) complexes (**1**–**3**). Rapid bacterial agglutination test in spiked water samples visualized by (**b**) the naked eye and (**c**) optical microscopy. In (**b**), 10^8^ CFU/mL MRSA and CRAB were treated with 400 μg/mL (or 420 μM) complex **1** and 400 μg/mL (or 320 μM) complex **2**. Labels: 1, only bacteria; 2, only complex **1**; 3, complex **1** + bacteria; 4, only complex **2**; 5, complex **2** + bacteria. In (**c**), 10^8^ CFU/mL of MRSA was treated with 400 μg/mL of complexes **1** and **2**, and results were observed in complementary bright-field (**i**–**iii**) and fluorescence (**iv**–**vi**) modes. Reproduced with permission from Ref. [114]. Copyright 2020, American Chemical Society.

**Figure 21 molecules-27-06440-f021:**
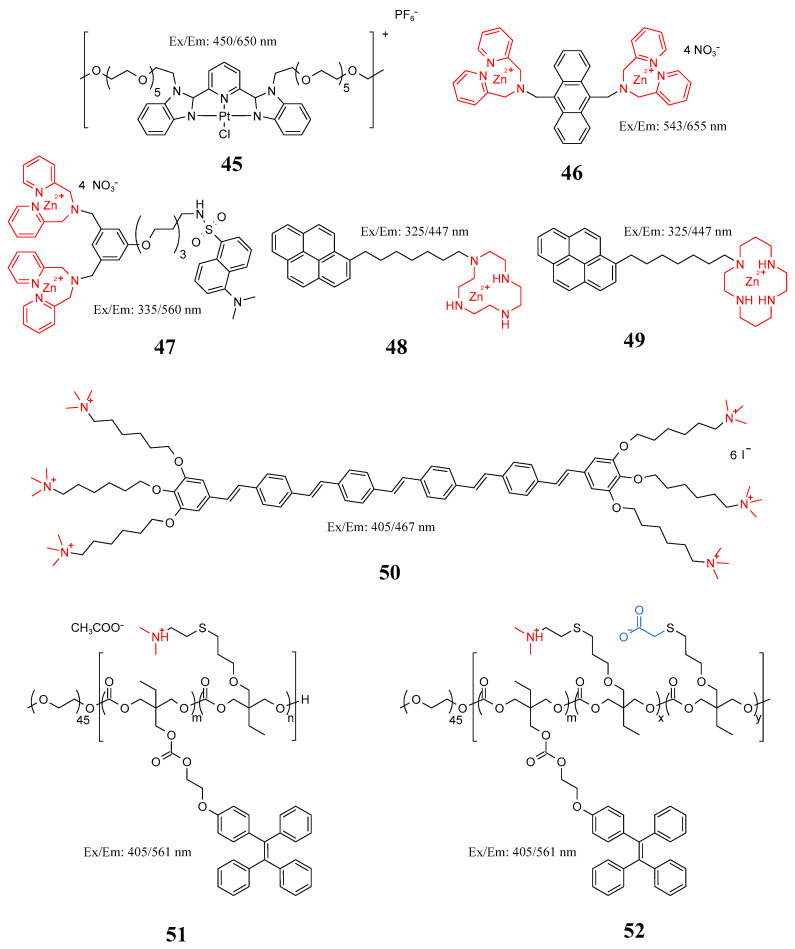
Structure of probes **45**–**52**.

**Figure 22 molecules-27-06440-f022:**
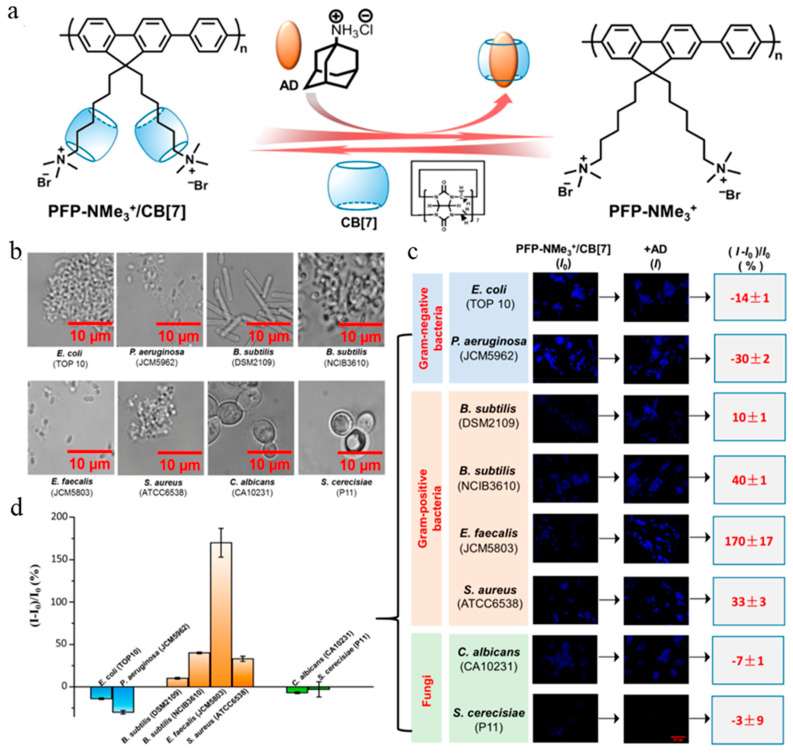
(**a**) Schematic supramolecular complex of PFP−NMe_3_^+^/CB[7] and reversible disassembly by AD. (**b**) Bright-field images of eight species of pathogens treated with PFP−NMe_3_^+^/CB[7] for 20 min. (**c**) Fluorescence images and intensity changes of PFP−NMe_3_^+^/CB[7] to the surface of eight pathogens before and after the addition of AD (scale bar is 20 μm). All microbe species were stained by blue fluorescence and treated with PFP−NMe_3_^+^/CB[7] for 10 min after the addition of AD, with an additional 10 min of interaction and an exposure time of 5 ms. The fluorescence intensity changes in parallel experimental groups were obtained using DVC View and Microsoft Excel software. PFP−NMe_3_^+^/CB[7] = 1:20, CB[7]: AD = 1:5, [PFP−NMe_3_^+^] = 15 μM in RUs. (**d**) Histogram of fluorescence intensity changes according to the data in presented in panel c. Reproduced with permission from Ref. [119]. Copyright 2018, American Chemical Society.

**Table 1 molecules-27-06440-t001:** WHO list of global priority pathogens.

Priority Category	Pathogen	Gram Stain	Antibiotic Resistance
Critical	*Acinetobacter baumannii*	-	Carbapenem-resistant
	*Pseudomonas aeruginosa*	-	Carbapenem-resistant
	*Enterobacteriaceae*	-	Carbapenem-resistant Third-generation cephalosporin-resistant
High	*Enterococcus faecium*	+	Vancomycin-resistant
	*Staphylococcus aureus*	+	Methicillin-resistant Vancomycin intermediate and resistant
	*Helicobacter pylori*	-	Clarithromycin-resistant
	*Campylobacter*	-	Fluoroquinolone-resistant
	*Salmonella spp.*	-	Fluoroquinolone-resistant
	*Neisseria gonorrhoeae*	-	Third-generation cephalosporin-resistant Fluoroquinolone-resistant
Medium	*Streptococcus pneumoniae*	+	Penicillin-non-susceptible
	*Haemophilus influenzae*	-	Ampicillin-resistant
	*Shigella spp.*	-	Fluoroquinolone-resistant

## Data Availability

Not applicable.

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
