# Peer review of "Recent Progress in Identifying Bacteria with Fluorescent Probes"

_molecules, 2022, doi:10.3390/molecules27196440_

Round 1

Reviewer 1 Report

In the manuscript, the authors summarized the recent progress on fluorescent probes for identifying bacteria based on varied recognition strategies. The Grammatical tense is disordered in this manuscript, some are past tense and others are present tense. Therefore, the manuscript needs to be grammar checked. Besides, the following concerns should be addressed.

1.     In the abstract and introduction Sections, the authors only focused on foodborne bacteria. However, “identifying bacteria” was used in the name of the review. Maybe the range of the name is too large.

2.     In the abstract Section, the authors mentioned “various fluorescent sensors (organic...metal ion probes)”, while only organic fluorescent probes were reviewed in the text. Similarly, the authors said “chromogenic elements include organic dyes, quantum dots, conjugated polymers, etc” and explained the advantages of them. However, only probes based on organic dyes were reviewed in detail. These are confusing to readers and the authors should reorganize and explain this.

3.     In Line 60, “quantum dots…consist of elements â…£, â…¡-â…¥, â…£-â…¥, or â…¢-â…¤”. What does “elements â…£, â…¡-â…¥, â…£-â…¥, or â…¢-â…¤” mean? I think such description is unclear.

4.     The uses of abbreviation in the manuscript are totally turbid. Actually, the word should be abbreviated when it appears for the first time and the corresponding abbreviation is used in the following manuscript. The authors should careffuly check the whole paper and revise such problems.

5.     In Line 257, “NH3+”. 3 should be subscript and + should be superscript. Similar error could be seen in Line 516.

6.     There is an error in Line 271 “LPS is a …Gram-positive bacteria”. In fact, LPS is the component of cell wall of Gram-negative bacteria.

7.     Figure 14 is unclear.

8.     The forms of references are non-uniform, including both the names of articles and journals. The authors must carefully check them one by one.

Reviewer 2 Report

This paper reviewed the recent progress in the discovery and development of fluorescent probes for the detection/identification of bacteria. The specific fluorescent probes for the bacteria detection/identification were highlighted. The topic fits the scope of Molecules and will benefit the discovery and development of novel bacterial detection/imaging methods, ultimately advancing the diagnosis and treatment of infection diseases. The manuscript is well-organized while some key issues should be addressed before its publication on Molecules.

Major points:

1. The structures of the represented fluorescent probes should be checked and corrected, as some structures (e.g., 6, 12, 16, etc) are not completed. One completed structure of ionic compound should contain both cation and anion.

2. There are lots of full names and abbreviations of bacteria strain names mixed together. The full names with abbreviations are required to be shown up at the first place, and then only abbreviations are needed for the others. The authors are required to check all.

3. The wavelengths of absorption/emission for the representative probes should be included in the manuscript if available.

4.  The detection/identification mechanisms of the representative probes should be illustrated with reaction schemes.

5. Pyridinium-containing compounds (Ref. Liu, et al. Journal of Materials Chemistry C, 2019, 7(40): 12509-12517.) as an important group of fluorescent probes, should be discussed in the section 3.3.

6. Caspase-1, as the host enzyme in the infection, should be discussed for the detection/identification of intracellular bacteria. (Representative ref.: Qi, et al. Angewandte Chemie International Edition, 2019, 58(45): 16229-16235.)

7. Hg (II) detection for the detection/identification of bacteria should be discussed. (Representative ref.: Pan, et al. Chemical Communications, 2018, 54(39): 4955-4958.)

8. The chloride detection for the detection/identification of bacteria should be discussed. (Representative ref.: Tutol, et al. Chemical science, 2021, 12(15): 5655-5663.)

Minor points:

1. In Figure 1, the related enzymes/substructures involving in the current development of fluorescent probes for bacteria detection/identification are suggested to be included for the better readability.

2. The resolution of Figure 5a, 13a, 14, 21d is too low and should be adjusted.

Reviewer 3 Report

In this review, Piao, Zhou and coworkers describe different techniques for the rapid and efficient fluorescent visualization of gram-positive and gram-negative bacteria. This is a comprehensive review that includes sections on chemosensors for the detection of metabolically labeled surface components, cell wall components, endogenous enzymes and nonspecific site of pathogenic bacteria. This topic is of interest for the Special Issue on Fluorescent probes. Bacteria resistance to antibiotics is a threat for human health and the use of fluorescence is a powerful tool for the instant detection of low quantities of a target analyte.

The present submission requires minor revisions before being considered for publication.

1) What is the desired detection limit for pathogenic bacteria? In the conclusion the authors mentioned the following “However, the high detection limit of this kind of probe still has some limitations in the field of bacterial detection.” Is the detection limit of the existing probes low enough for practical applications? Furthermore, could the authors comment on the selectivity of these probes for pathogenic bacteria in the presence of non-pathogenic bacteria.

2) The following recent reviews on this topic should be included.

a. Yoon, J. et al. Recent progress in fluorescent probes for bacteria, Chem. Soc. Rev., 2021, 50, 7725-7744. DOI: 10.1039/D0CS01340D.

b. Xu, Z. et al. Development of fluorescent probes targeting the cell wall of pathogenic bacteria, Coord. Chem. Rev., 429, 213646. DOI: 10.1016/j.ccr.2020.213646

c. Wang, Z. et al. Small-molecule fluorescent probes: big future for specific bacterial labeling and infection detection, Chem. Commun. 2022, 58, 155. DOI: 10.1039/d1cc05531c

3) All figures must be referred in the main text. For instance, Figure 19 and 21 are not referred in the main text.

4) The quality of the ChemDraw figures is excellent, however the quality of the reprinted figures is poor and should be improved. In Figure 14, the figure captions a, b and c are indicated but are not present in the figure, furthermore the structure of P4-FITC is blurred. In Figures 5, 13 and 19, the presence of numbers for the compounds is confusing since these numbers are already attributed to other compounds. (For instance, compounds 1-6 in Figure 5 are not the same as 1-6 in Figure 2).

5) The writing in the manuscript will need some improvement to the standards of publication, especially for some problems with spelling, grammar and style.

The following sentences are a few examples of sentences that need to be reformulated.

Lines 118-120, p. 4 “Mycobacterium and other members of the Corynebacterium family surface mycomembrane (MM) are special outer membranes that act as the main defensive barrier, endowing M. tuberculosis and related bacteria with drug resistance [40]”

Line 130, p. 5: “MM consists of lipid and glycolipid, which are wound in a waxy coating between arabino galactose and surface oligosaccharide layers.”

Lines 146-148, p. 5: “The molecular weight, hydrophobicity, and charge of the probe have an effect on passive diffusion or through outer membrane pores, so FDAAs have some limitations in labeling Gram-negative bacteria and Mycobacterium [47].”

Lines 264-266, p. 10 :“However, probe alkalinity morpholine groups with the LTA acidic structure of Gram-positive bacteria are compatible but don’t fit with the outer structure of the fungus, making it easier for the insert part has been removed, thus specific identification of Gram-positive bacteria [77].”

Lines 366-367, p. 14: “Yang et al. combined AgNCs with bacterial aptamers to achieve a new method for detecting and killing bacteria with the naked eye”

Line 377, p. 14: “The metabolic activity of life cannot leave enzyme catalysis.”

Lines 406-408, p. 15 : “Gwynne et al. Synthetic probe 30 detects alkaline phosphatase in bacteria and is selective for S. aureus compared to Enterococcus faecalis, pseudomonas aeruginosa, and E. coli [105].”

Lines 417-420, p 15. “Ma et al. synthesized NIR fluorescence 31 by adding cephalosporin into the Semi-cyanine skeleton.” should be replaced by “Ma et al. synthesized NIR fluorescent probe 31 by adding cephalosporin into the Semi-cyanine skeleton.”

Line 437, p. 16 “Therefore, I will introduce” should be reformulated

Line 513-515, p. 18: “Zhou et al. synthesized conjugated oligoelectrolytes to probe 48, and the blocking of the outer membrane of Gram-negative bacteria made the probe specific to recognize Gram-positive bacteria [124].”

Additional minor corrections:

Line 120, p. 4: “As figure 3 shown” should be replaced by “As shown in Figure 3”

Lines 142-143, p.5: “4-N, n-dimethylamino1, and 8-Naphthalimide” should be replaced by “4-N,N-Dimethylamino-1,8-naphthalimide”

Line 204, p. 8: “As figure 5 shown” should be replaced by “As shown in Figure 5”

Lines 250-252, p.10: In the sentence “Teichoic acid is a unique component of the cell wall of Gram-positive bacteria. It is a weakly acidic substance made of ribitol or glycerol residues linked to each other by phosphoric acid double bond”, “phosphoric acid double bond” should be replaced by “phosphate groups”.

Line 302, p. 12: Probe 23, was not designed by Tsuchido but by Hu et al. (ref 85)

Line 336, p 13. “1-bromothalene” should be replace by “1-bromonaphthalene”

Lines 427-429, 16: In the sentence “Mehta et al. conjugated an environmentally sensitive fluorophore onto a mercaptan-based scaffold of New Delhi metal-β lactamase inhibitors, probe 33, for monitoring the dynamic metallization of NDM in E. coli and for fluorescence imaging [109].” “New Delhi metal-β lactamase” should be replaced by “New Delhi metallo-β-lactamase (NDM)” and “metallization” should be “metallation”.

Line 454, p.17: TBP is not defined.

Line 473, p. 17: “luminescence life” should be replaced by “luminescence lifetime”

Line 476, p. 17: “synthesis route” should be replaced by “synthetic route”

Line 480, p.17: “Gupta developed” should be replaced by “Gupta et al. developed”.

Line 485, p.18: “2,6 bis (benzimidazole 20-yl) pyridinium with hexagon methyl ether” should be replaced by “2,6-bis(benzimidazol-2′-yl)pyridine with hexaethylene glycol methyl ether”

Line 505, p. 19: There is no compound 49 in Figure 20. Please correct the Figure caption.

Round 2

Reviewer 1 Report

Accept

Reviewer 2 Report

After the authors’ revision according to my and others’ previous comments, the quality of this paper was significantly improved, and could reach the required quality standard for Molecules in my opinion. I suggest accepting it without further revision.